# *Aspergillus* Conidia and Allergens in Outdoor Environment: A Health Hazard?

**DOI:** 10.3390/jof9060624

**Published:** 2023-05-28

**Authors:** Concepción De Linares, David Navarro, Rut Puigdemunt, Jordina Belmonte

**Affiliations:** 1Department of Botany, University of Granada, 18071 Granada, Spain; 2Departament de Biologia Animal, Biologia Vegetal i Ecologia, Universitat Autònoma de Barcelona (UAB), 08193 Bellaterra, Spain; david.navarro@uab.cat (D.N.); rut.puigdemunt@uab.cat (R.P.); jordina.belmonte@uab.cat (J.B.); 3Institut de Ciència i Tecnologia Ambientals (ICTA-UAB), Universitat Autònoma de Barcelona (UAB), 08193 Bellaterra, Spain

**Keywords:** aerobiology, Asp f 1, ELISA, fungal spores, high volume sampler, hirst sampler

## Abstract

*Aspergillus* is a genus of saprophytic fungus widely distributed in the environment and associated with soil, decaying vegetation, or seeds. However, some species, such as *A. fumigatus*, are considered opportunistic pathogens in humans. Their conidia (asexual spores) and mycelia are associated with clinical diseases known as invasive aspergillosis (IA), mainly related to the respiratory tract, such as allergic asthma, allergic bronchopulmonary aspergillosis (ABPA), or hypersensitivity. However, they can also disseminate to other organs, particularly the central nervous system. Due to the dispersal mechanism of the conidia through the air, airborne fungal particle measurement should be used to prevent and control this mold. This study aims to measure the outdoor airborne concentration of *Aspergillus* conidia and the Asp f 1 allergen concentration in Bellaterra (Barcelona, Spain) during 2021 and 2022, and to compare their dynamics to improve the understanding of the biology of this genus and contribute to a better diagnosis, prevention, and therapeutic measures in the face of possible health problems. The results show that both particles were airborne nearly all year round, but their concentrations showed no correlation. Due to Asp f 1 not being present in the conidia itself but being detectable during their germination and in hyphal fragments, we report the relevance of the aero-immunological analysis as a methodology to detect the potential pathogenic hazard of this fungus.

## 1. Introduction

Airborne fungal particles are an important essential fraction of atmospheric indoor and outdoor components that can be released by active or passive processes [1]. In general, fungi release large amounts of spores (sexual or asexual) and hyphal fragments into the air, which can be involved in the decomposition of food and raw materials or the deterioration of organic matter during storage [2]. However, airborne fungi cause serious health problems, such as induced inflammatory responses or respiratory tract infections [3,4]. For example, in Europe, the incidence of fungus-induced respiratory tract allergies is as high as 20–30% in the atopic population, reaching 6% in the general population [5]. 

More than 112 fungi are recognized as allergenic sources [6]. One of the most important is *Aspergillus*, considered the major primary fungal causative agent of allergic bronchopulmonary mycosis (ABPM) and respiratory allergic symptoms [7,8]. This Ascomycota is a filamentous saprophytic fungus widely distributed and associated with soil, decaying vegetation, or seeds. However, some species, such as *A. fumigatus*, are considered opportunistic pathogens of immunocompromised hosts. Their conidia (asexual spores) and mycelia are associated with clinical diseases related to the respiratory tract, such as allergic asthma, ABPM, or hypersensitivity, collectively termed ‘aspergillosis’ [9]. 

There are currently defined five major molecular allergens of *A. fumigatus* (Asp f 1, Asp f 2, Asp f 3, Asp f 4, and Asp f 6). Among them, Asp f 1 is the most important. Its production is related to the germination of spores and growth of the mycelia’s early fungal invasion [10], which is related to fungal colonization and the saprophytic nature of the fungi.

Nowadays, there is no universal, recommended method for continuous air sampling that can accurately show the entire aeromycobiota of a location [11]. The volumetric methods based on Hirst-type samplers offer a continuous sampling from the air, with the reliable determination of fungal spores. However, it is less effective in the case of particles with a diameter below 5 µm, as occurs with *Aspergillus*, *Penicillium*, and some basidiospores, which may lead to the underestimation of the number of spores in the air [11]. Another problem is the identification of fungal spores. In the case of *Aspergillus* and *Penicillium*, the conidia of both fungi genera present similar morphology and are indistinguishable under a light microscope. Therefore, in aerobiological studies, both fungi are grouped together as *Aspergillus/Penicillium* spore type [12]. Moreover, this methodology cannot identify smaller fungal fragments and hyphae [13]. The viable methods (cultivation on the medium) can resolve the identification of these genera; they even can identify the order of the species [14]. However, continuous samples are not possible. In contrast to these traditional methods, the immunological test may detect the presence of fungi based on the detection of their major allergen. Although this methodology depends on the commercial availability of specific allergens [13], it could be an alternative method for continuous airborne control of these fungi. 

This study compares the airborne dynamics of *Aspergillus/Penicillium* spore type and the concentration of Asp f 1 allergen in order to contribute to better diagnoses, prevention, and therapeutic measures for aspergillosis patients.

## 2. Materials and Methods

### 2.1. Sampling Site

The aerobiological station is located at the Universitat Autònoma de Barcelona (Bellaterra, Barcelona, NE, Spain), on the rooftop of the C building, at 23 m.a.g.l. (41°30′20″ N, 02°06′28″ E and 245 m.a.s.l.). The climate of this locality can be described as fresh (mean annual temperature 12.0–15.5 °C) and humid (total annual precipitation 400–700 mm), and the corresponding phytoclimate is Fresh-Continental Oriental-semihumid [15]. 

### 2.2. Aerobiological Methodology

A Hirst-type volumetric spore trap (VPPS 2000 spore trap, Lanzoni s.r.l.) was used for the aerobiological fungal spore study. This collector aspires at a flow rate of 10 L/min, comparable with the respiration of an average human adult, and was designed specifically for the intake of spores, pollen, and other particles suspended in the air [16]. It is a continuous volumetric sampler with wind orientation of the intake orifice, and, thanks to an aspiration system, the collected particles are deposited on a plastic band coated with silicone oil tightly deposited around a drum that is displaced at a rate of 2 mm/hour using a clockwise mechanism. Once a week, the plastic band in the drum is replaced, and the segments corresponding to each day (0–24 h UTC) are mounted on a slide that is analyzed at the light microscope. The spore monitoring was performed continuously between 1 January 2021 and 31 December 2022 under the Spanish Aerobiological Network (Red Española de Aerobiología, REA) [17] and following the minimum requirements of the European Aerobiology Society, EAS [18]. The terminology used in this paper follows the International Association for Aerobiology (IAA) and EAS recommendations [19]. The data used in this study were expressed as the daily average of spores per cubic meter of air (spores/m^3^) and Annual Spores Integral (ASIn, spore * day/m^3^), or the addition of the average daily spores’ concentration over the year. The fungal taxa considered were the *Aspergillus/Penicillium* spore type. In both cases, the asexual spores (conidia) extend out in long chains from the vegetative mycelium and readily detach to become airborne [20]. The size of this spore type is around 2.5 µm in diameter, with a spherical morphology that may have faint cinereous pigmentation or be hyaline. On their surface appear slight ornamentation similar to minute dots at the perimeter. Under a light microscope, it is frequently observed that the spores are in short chains or small groups adhered together (Figure 1). Due to the difficulties in the differentiation of this morphology and because the conidia may be hidden along with other airborne particles, it is very plausible that they are underestimated.

Added to the information generated during this study, the daily concentrations of airborne *Aspergillus/Penicillium* spore type in the Bellaterra sampling site for the ten years (2011–2020) were used to analyze the representativity of the results obtained for years 2021 and 2022. These data were extracted from the database of the Catalan Aerobiological Network (Xarxa Aerobiològica de Catalunya, XAC) and are summarized in Table 1.

### 2.3. Immunochemical Methodology

Besides the Hirst sampler, the Asp f 1 sampling was performed during 2021 and 2022, using a high-volume sampler (MCV CAV-A/mb, ©MCV) designed in principle for the sampling of atmospheric particulate matter. It is a volumetric sampler, working at a flow intake rate of 400 L/min. and impelling the air against a fiberglass filter (UNE-EN12341:2015; UNE-EN 14907:2006a, b) that will retain the particles. Based on MCV S.L. Company, this collector complies with the specifications for high-volume collectors in the UNE-EN 12,341 and UNE-EN 14,907 standards. The total diameter of the filter is 15 cm, but the diameter of the impacted area is 12 cm. The daily samples were carried out from 0 to 24 h, and the sampling was carried out on alternate days to permit manual replacement. The samples were stored at 2 °C until their analysis. Before the analysis, a representative part of the daily filters, consisting of 8 circles of 0.5 cm in diameter obtained in selected areas, was equally distributed in two Eppendorf vials with 500 µL PBS solution and hydrated for 6 h [21]. Once hydrated, the samples obtained were analyzed according to commercial ELISA kits to detect Asp f 1 (Inbio, Indoor Biotechnologies, Inc, Cardiff, United Kingdom) according to the protocol provided on the supplier’s website [22]. The allergen standards used were provided with the kits. The microplates used were NUNC™ microplates. Absorbance measurements were made at a 492 nm filter in a plate reader model Halo MpR-96 from Dynamica^®^. The results were expressed as daily allergen concentrations in nanograms of allergen per cubic meter of air (ng/m^3^) and as Annual Allergen Integral (AAIn) obtained by summing the average daily allergen concentration over the year (ng * day/m^3^).

### 2.4. Meteorological Data

The meteorological variables considered in this study were maximum and minimum daily temperatures (expressed in degrees Celsius), daily rainfall (expressed in millimeters), and relative humidity (expressed in percentages). Data were obtained from the Servei Meteorològic de Catalunya and correspond to the Sabadell meteorological station (41°33′56″ N, 02°04′10″ E, 258 m.a.s.l.). 

### 2.5. Statistical Analysis

Spearman’s correlation coefficients were calculated between the daily concentrations of *Aspergillus/Penicillium* spore type and Asp f 1 allergen and daily meteorological variables. The analyses were carried out by using the IBM SPSS Statistic version 25.

## 3. Results

### 3.1. Aerobiological and Meteorological Results

Based on the data of the 10-year period previous to the years of this study, Aspergillus/Penicillium spore type accounted for 0.52% of all monitored fungal spores in the year (varying from 0.86% to 0.34%, Table 1). During the years 2021 and 2022, Aspergillus/Penicillium spore type showed a higher proportion with regard to the ASIn of all the spore types registered, accounting for 0.79 and 0.75%, respectively (Table 2). 

The dynamics along the year of the mean daily airborne Aspergillus/Penicillium spore type for the period 2011–2020 and the daily concentrations during 2021 and 2022 were studied (Figure 2). As can be observed, considering the mean values of the 10-year period results in this fungal spore type being present in the atmosphere for all 365 days of the year, while this is not the case in any of the years analyzed (Table 1 and Table 2, Figure 2). Moreover, the mean sporulation tended to be more important in autumn. An important aspect to consider is that when daily averages for the ten-year period are plotted, the concentrations become smoother, and important peak values, as the one for 31^st^ October 2013 (918 s/m^3^), are decreased (in this case to approximately 150 spores/m^3^). In relation to the years analyzed in this study (2021–2022), this spore type followed similar dynamics than in the previous 10-year period, although in the study period, the peak of the years was detected in May (23rd and 28th, respectively), and they varied from March to November (occurring each month with the only exception of July and August) between 2011 and 2020 (Figure 2). Therefore, we assumed that the results for the period studied can be considered comparable and be extrapolated to other years. 

The aerobiological parameters studied for Aspergillus/Penicillium spore type, Asp f 1, and for meteorological data showed that the aerobiological behavior of this spore type was similar in the years 2021 and 2022 (Table 2). In both cases, ASIn did not reach 12,000 spores * day/m^3^, the peak was similar (319 and 386 spore/m^3^), the peak dates were observed in spring with five days of difference, and the number of days of the presence of this spore type in the air was 237 days (over 364) in 2021 and 235 (over 356) in 2022. 

As for Asp f 1, it was also detected at similar levels in both years, with AAIn values of 507 and 505 ng * day/m^3^, respectively. The peak day in both cases occurred in February (23rd and 28th), and the presence of Asp f 1 was continuous for all the days studied. 

In relation to the meteorological parameters, data from the study period showed slightly higher temperatures in the year 2022, and much lower precipitation in both years than in the period 2011–2020. However, between 2021 and 2022, the values reached were similar without significant differences.

### 3.2. Airborne Fungal Aeroallergens vs. Spores and vs. Meteorology

Throughout the whole period studied, the annual patterns of the daily concentrations of the airborne Aspergillus/Penicillium spore type and Asp f 1 in 2021 and 2022 were present in the air an important number of days per year (over 65% of analyzed days), showing its maximum sporulation in spring and autumn (Figure 3). In relation to the dynamics of the aeroallergen Asp f 1, it was characterized as being present on all the days analyzed (alternate days) in 2021 and 2022, although with abundant oscillations in the concentration values. In both years, when airborne spores registered the highest concentrations, the values of the aeroallergen were lower. Moreover, in both years, the maximum airborne allergen concentrations were registered in winter (mainly in January and February, with the annual absolute peaks occurring in February).

The results of the Spearman correlation test applied to the daily values of Aspergillus/Penicillium spore type, Asp f 1 allergen, and the meteorological parameters showed no influence between them (Table 3). No significant correlations were found in the period studied. Only in 2021 did the spore type show a positive and significant correlation (*p* < 0.01) with precipitation and relative humidity. In the case of Asp f 1 allergen, both years showed a negative and significant correlation (*p* < 0.05; *p* < 0.01) with temperature. 

## 4. Discussion

The study of the daily dynamics of the *Aspergillus/Penicillium* spore type along the ten years previous to the analysis of Asp f 1 (Table 1, Figure 2) evidenced that it does not follow a clear seasonal pattern, that the spores are observed in the aerobiological plates a critical number of days per year (between 50% and 72% of the days analyzed per year) but not continuously, and that they are distributed over several months. However, the maximum airborne spore concentration period in autumn (October and November) can be intuitively established. This yearly phenomenon has been reported by other studies, such as Oliveira et al. [23] in Portugal and Rosas et al. [24] in Mexico. However, the dynamics of this spore type around the world are variable. Several authors have shown that the main period occurs during spring, as in Central Spain, Salamanca [25]. In contrast, others reported a two-season occurrence (spring and autumn), as in Poland [11] or Ireland [26]. In the USA [27] and Finland [28], the main period was observed during summer. Finally, some studies demonstrated that the *Aspergillus/Penicillium* spore type is independent of seasonal changes, such as in Denmark [29]. The correlation study between *Aspergillus/Penicillium* spore type and meteorological variables showed positive and significant correlations with precipitation and relative humidity. So, their sporulation could be favored by the rainfall and increasing humidity, as has occurred in other studies [11,23].

Although the period of high fungal spore concentrations does not show a unique pattern, it is indisputable that the existence of this spore type in the air has implications for patients with problems in the respiratory system [20]. Even so, there is no general agreement on analyzing the presence of airborne fungal particles in the atmosphere. Some authors analyzed the air using viable sampling methods that resulted in the calculation of colony-forming units (CFU), which means brief periods of sampling sporadically through the year [30,31,32]. Other authors use aerobiological methods based on palynology (with the identification of the particles through an optical microscope) [23,25]. In this case, the air sampling is continuous, but there is no consensus on the surface of the sample analyzed. These aerobiological studies have a substantial limitation in counting and identifying the numerous and diverse fungal spore types present in the atmosphere, especially in the case of *Aspergillus/Penicillium* spore type, due to their small size and hyaline color. Furthermore, this methodology cannot identify smaller fungal fragments or hyphae, considered a significant source of allergens [33], so the actual allergenic risk due to *Aspergillus* would still not be observed.

In recent decades, aerobiological studies have evolved, incorporating molecular analytical methods, such as detecting and quantifying the allergen load in the atmosphere through immunological techniques. This new methodology allows the comparison of their results with those from traditional sampling, achieving a better knowledge of the natural aerobiological dynamics of pollen, spores, and allergens. Therefore, the information obtained with both methodologies can help assess pollen/spore exposure and improve research on the epidemiology and clinical trials. The present study offers an analysis of the airborne *Aspergillus/Penicillium* spore type and the allergenic load of Asp f 1 in Bellaterra (Barcelona) to determine if the traditional aerobiological method reflects the real risk period of Asp f 1 in the air.

This study recorded airborne dynamics of *Aspergillus/Penicillium* spore type in 2021 and 2022. As shown in our results and other research [34,35], the levels of this spore type can be variable throughout the year, with high values in some seasons and low values in others. During the period studied, high daily concentrations of spores were registered on certain days of the year exceeding 200 spores/m^3^. While in 2021, this occurred on some days of April, May, and June, in 2022, this was registered in May, June, and October. This fact, and the lack of correlation with meteorological variables, could be explained by the sporadic behavior of these spores [11]. However, the positive and significant correlation between the two years studied (0.145; *p* < 0.01) shows how this spore type presents similar annual airborne dynamics, although some days randomly register high concentrations.

In this study, the authors used a high-volume sampler with a suction capacity of 400 L/min. The election of this sampler was based on the hypothesis that this high-volume collector could collect biological particles, such as allergens, at the same time as inorganic particles. As shown in De Linares et al. [21], sample collection analysis and quantification of fungal allergens are possible with this sampler and using the ELISA technique.

Despite several studies on airborne allergens, they are mainly focused on allergens from pollen, and there need to be more studies on fungal spores. The research of aeroallergens from fungal spores was mainly based on the detection and quantification of Alt a 1 [21,36,37], because it is the most critical fungal aeroallergen that provokes respiratory tract symptoms. Studies related to outdoor *Aspergillus* spore and their major allergen (Asp f 1) are scarce. In this case, we found one article using a similar methodology: a volumetric sampler with a flow rate of 40 L/min [38] to quantify the airborne Asp f 1. As in this study, we found that the seasonal distribution of airborne spores did not correlate with the aeroallergen load. In our case, during the two-year study (2021–2022), the higher airborne spores’ concentrations were registered in spring and autumn, with the peak days on 23 May and 28 May, respectively. The fact that the maximum spore concentrations were registered in the more humid season agrees with other studies [11,34,35,38,39]. However, the maximum concentration of the Asp f 1 allergen was found during winter, peaking on 18 February and 11 February, as occurred in Vermani et al. [33], which detected the highest outdoor Asp f 1 concentration of *A. fumigatus* and *A. niger* in February too.

Statistical results showed no significant correlation between meteorology and *Aspergillus/Penicillium* spore type. Only in 2021 were positive correlations registered with precipitation and relative humidity. The same lack of correlation with temperature was obtained by Grinn-Gofron [11], Sadys et al. [40], and Akgül et al. [41]. In the case of Asp f 1 allergen, non-correlation was detected in any year, so it could indicate that the weather conditions do not influence these small particles and depend on other mechanisms that must be studied.

Consequently, based on the discrepancies in the moments of these two particles being present in the air and the similitudes in the date of maximum concentrations in both years and with other studies, we postulate that

(a)The coincidence of the values of ASIn, AAIn, and the dates of the peaks of spores and allergens, as well as the dynamic found during the two years studied, confirm that the detection of the *Aspergillus/Penicillium* spore type and Asp f 1 allergen is possible with both aerobiological techniques, and follow a defined behavior along the time;(b)If we accept the postulate of Sporik et al. [30] that Asp f 1 is not present in the spore but in hyphae fragments, and as previous studies have shown, we accept too that the level of the fungal hyphae can be relatively high in the air [42];(c)*Aspergillus/Penicillium* spore type is conditioned to precipitation and relative humidity, which promote the production and release of the spores of these fungi. On the other hand, the aeroallergens registered in winter could be derived from hyphal fragments, which, under the light microscope, are extremely difficult to distinguish but can be detected through the allergens with immunological techniques.

## 5. Conclusions

Asp f 1 daily concentration measurements have been obtained, for the first time in Catalonia, at the Bellaterra (Barcelona) aerobiological station, for the years 2021 and 2022, showing that the MCV CAV-A/mb high-volume sampler has proven to be valuable and sensitive enough to the detection and quantification of the allergen Asp f 1 using ELISA Kits.

Despite being unable to explain specific temporal differences observed between concentrations of spores and allergens, we recommend using these two methodologies (quantification of fungal spores under a light microscope and immunochemical quantification to detect allergens) to detect, control, prevent, and diagnose this fungal taxon. Further studies must focus on establishing the origin of the aeroallergen registered in periods without airborne fungal spores and determine which moment the exposure is more hazardous when there are high concentrations of spores or the allergen.

## Figures and Tables

**Figure 1 jof-09-00624-f001:**
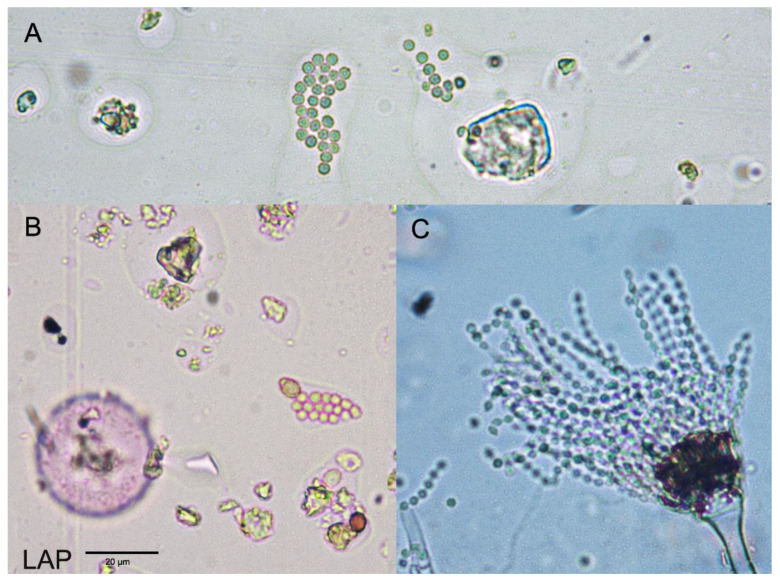
*Aspergillus/Penicillium* spore type. (**A**) Spores observed on aerobiological samples; (**B**) comparative image between the size of *Aspergillus/Penicillium* spores and Cupressaceae pollen; (**C**) conidiophore with conidiospores of *A. fumigatus*.

**Figure 2 jof-09-00624-f002:**
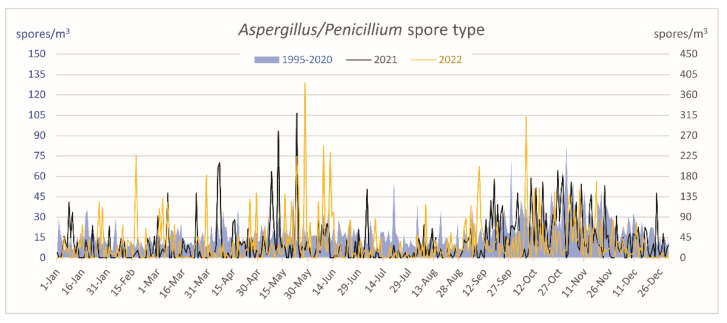
Airborne *Aspergillus/Penicillium* spore type daily dynamics along the year. Mean data of the period 2011–2020 (blue area and left *y*-axis) vs. 2021 (black line and right *y*-axis) and 2022 (yellow line and right *y*-axis).

**Figure 3 jof-09-00624-f003:**
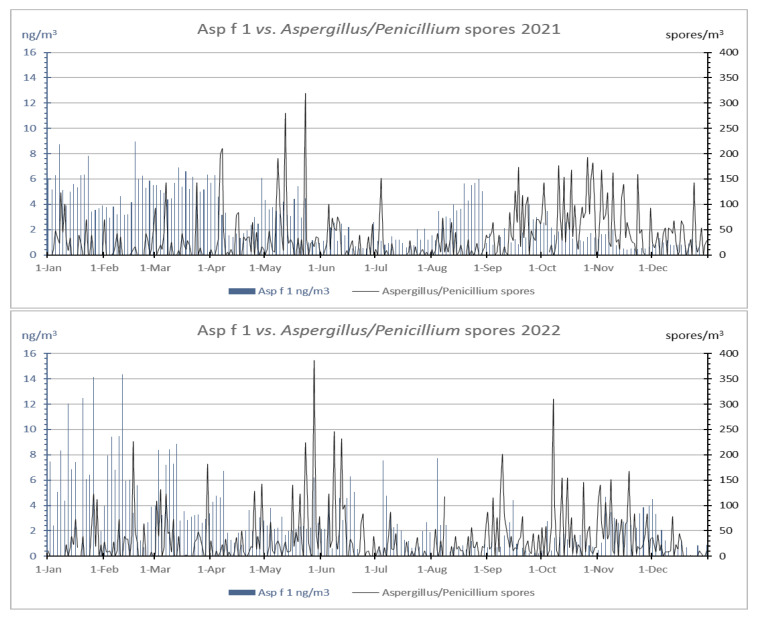
Daily dynamics of airborne *Aspergillus/Penicillium* spore type and Asp f 1 allergen during 2021 and 2022.

**Table 1 jof-09-00624-t001:** Main aerobiological parameters of *Aspergillus/Penicillium* spore type for the period 2011–2020 (data from the Xarxa Aerobiològica de Catalunya-XAC) database and summary of the main meteorological parameters. ASIn: Annual Spore Indice; ATSIn: Annual Total Spore Indice; T: temperature; P: annual rainfall; RH: related humidity.

	2011	2012	2013	2014	2015	2016	2017	2018	2019	2020	Mean 2011–2020
ASIn (spore * day/m^3^)	16,106	9730	8462	11,421	9817	8660	6964	9724	6759	8812	9645
Peak (spores/m^3^)	624	591	918	395	361	252	207	392	274	708	472
Peak day	21-nov	28-sep	31-oct	26-nov	09-apr	13-mar	04-oct	17-jun	29-may	05-apr	--
Nr of days with spores	222	190	178	246	259	249	237	250	227	209	227
Nr of analyzed days	364	364	357	362	359	360	361	364	361	340	359
ATSIN (spore * day/m^3^)	1,873,455	2,065,498	1,986,474	2,567,732	1,427,132	1,898,641	1,401,089	2,867,906	1,259,152	1,644,947	1,899,202
% vs. Total Spores	0.86	0.47	0.43	0.44	0.69	0.46	0.50	0.34	0.54	0.54	0.52

T Max (°C)	21.1	21.0	20.6	21.2	21.8	21.5	21.7	21.2	21.7	21.4	21.3
T Min (°C)	9.6	8.9	8.8	9.7	9.7	9.7	9.4	9.9	9.3	9.6	9.5
T Mean (°C)	15.3	15.0	147	15.5	15.7	15.6	15.6	15.6	15.5	15.5	15.4
P 0–24 h (mm)	809.3	445.6	563.7	639.0	326.6	441.6	468.8	919.4	553.0	847.2	601.4
RH (%)	74.2	69.6	71.8	74.4	68.9	69.7	68.8	73.3	68.6	73.8	71.3

*: daily spores per cubic meter.

**Table 2 jof-09-00624-t002:** Main aerobiological parameters of *Aspergillus/Penicillium* spore type and Asp f 1 allergen and main meteorological conditions in Bellaterra, Barcelona, Spain, during the period studied. T: temperature; P: annual rainfall; and RH: relative humidity.

		2021	2022
Hirst sampler	*Aspergillus/Penicillium* spores. ASIn (spore * day/m^3^)	10,912	9957
	Peak (spore/m^3^)	319	386
	Peak day	23-may	28-may
	Nr of days with *Aspergillus/Penicillium* spore type	237	235
	Nr of analyzed days	364	356
	Total spores. ASIn (spore * day/m^3^)	1,380,260	1,329,525
	% vs. Total spores	0.79	0.75
High-volume sampler	Asp f 1. AAIn (ng * day/m^3^)	506,7	505,0
	Peak (ng/m^3^)	9.0	14.4
	Peak day	18-feb	11-feb
	Nr of days with Asp f 1	179	163
	Nr of analyzed days	179	163
Meteorological data	T Max (°C)	21.1	22.7
	T Min (°C)	9.4	10.3
	T Mean (°C)	15.3	16.5
	P 0–24h (mm)	417.9	393.7
	RH (%)	72.6	70.6

*: daily spores per cubic meter.

**Table 3 jof-09-00624-t003:** Correlation coefficients for *Aspergillus/Penicillium* spore type, Asp f 1, and meteorological variables.

	2021				2022			
	Asp./Pen. Spore Type		Asp f 1		Asp./Pen. Spore Type		Asp f 1	
	Spearman T.	N	Spearman T.	N	Spearman T.	N	Spearman T.	N
**Asp./Pen. spore type**	1	367	−0.098	181	1	359	−0.008	163
**Asp f 1**	−0.098	181	1	182	−0.008	163	1	166
**TMax (°C)**	−0.043	367	**−0.186 ***	182	0.057	359	**−0.348 ****	166
**TMin (°C)**	−0.021	367	**−0.207 ****	182	0.043	359	**−0.408 ****	166
**TMean (°C)**	−0.033	367	**−0.189 ***	182	0.050	359	**−0.409 ****	166
**P 0–24 (mm)**	**0.234 ****	367	0.040	182	0.039	359	0.061	166
**RH (%)**	**0.182 ****	367	0.104	182	0.101	359	−0.067	166
					Spearman T.	N		
**Asp./Pen. spore type (2021) vs. Asp./Pen. spore type (2022)**					**0.145 ****	358		
**Asp f 1 (2021) vs. Asp f 1 (2022)**					**−0.022**	31		

** *p* < 0.01; * *p* < 0.05. The blue and red shadows and bold format indicate the more relevant results. (Asp./Pen. = *Aspergillus/Penicillium*).

## Data Availability

The data presented in this study are available on request from the corresponding author. The data are not publicly available due to privacy.

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
