# Peer review of "Aspergillus Conidia and Allergens in Outdoor Environment: A Health Hazard?"

_jof, 2023, doi:10.3390/jof9060624_

Round 1

Reviewer 1 Report

Review of Aspergillus Condida and Allergens in Outdoor Environment

Aspergillus is a common indoor and outdoor fungi which can cause asthma, infection and mycotoxin exposure.  This study is an interesting study of outdoor Aspergillus/Penicillium levels and asp f1 allergen.   A strength of this study is that it involved daily concentrations of Aspergillus/ Penicillium over  a 10 year period, and 2 years of sampling for the Aspergillus allergen Asp f1.   I think this will be a useful addition to the literature.  I have some comments which might be helpful for improving the paper.

TOTAL FUNGI.  What were the levels of total fungi sampled in this syudy?  If this data is available it might be interesting to include

OTHER COMMON OUTDOOR FUNGI AND THEIR ALLERGENS

Have you considered doing further studies on other common fungi as well as their allergens?

FIGURE 1- What species of Aspergillus is present on part C- perhaps Aspergillus fumigatus ___ ????    I know from my own experience that photographing fungi is difficult- but perhaps could the contrast of these slides be increased?

PEAK LEVELS OF ASPERGILLUS/ PENICILLIUM FROM 2011 TO 2020 IS ONLY 918 SPORES/M3.  Any commentary why these peaks are all below 1,000 spores/ cubic meter?  During peak periods- spores from these 2 species often exceed 1,000 spores cubic meter.

Seems Fairly Good

Author Response

We want to thank the reviewer for the work done, which helps us to improve the manuscript. We have taken the proposals into account. You can see it in the manuscript with the changes marked and in the responses to reviewer 1.pdf.

Reviewer 2 Report

Title: Aspergillus conidia and allergens in outdoor environment, a health hazard?

Reference: jof-2345681

Type: Research

Overview:

The manuscript jof-2345681 entitled “Aspergillus conidia and allergens in outdoor environment, a health hazard?” aims to measure the outdoor airborne concentration of Aspergillus conidia and Asp f 1 allergen, during 2021 and 2022, and to compare their dynamics to understand the biology of this genus and contribute to a better diagnoses, prevention, and therapeutic measures in front of possible health problems.

The manuscript was checked for plagiarism and AI content.

General comments:

Overall, the manuscript well-written. However, minor changes are required.

Abstract and Keywords: improve English style and grammar (as suggested in the following sections). Sections underlined correspond to plagiarism detection. Present keywords in alphabetic order.

The Introduction section is clear and sufficient. However, the authors should improve English style and grammar (as suggested in the following sections).

The Material and Method section: high percentage of plagiarism detected (nearly 50%). A version including English corrections is presented. Due to the need to be re-written this section, the changes were not signaled.

The Result section: needs to be re-written. For instance, avoid using sentences such as “Figure 2 shows”. Describe the results and them indicate the figure. This section was not reviewed.

The Discussion section needs simplification.

The Conclusion section is clear and sufficient.

Specific comments:

Abstract: Please consider replacing with:

Abstract: Aspergillus is a genus of saprophytic fungus widely distributed in the environment and associated with soil, decaying vegetation, or seeds. However, some species, such as A. fumigatus, are considered opportunistic pathogens in humans. Their conidia (asexual spores) and mycelia are associated with clinical diseases known as invasive aspergillosis (IA), mainly related to the respiratory tract, such as allergic asthma, allergic bronchopulmonary aspergillosis (ABPA), and hypersensitivity, etc., but. However, they can also disseminate to other organs, particularly the central nervous system. Due to the dispersal mechanism of the conidia through the air, airborne fungal particles measurement should be used to prevent and control this mold. This study aims to measure the outdoor airborne concentration of Aspergillus conidia and the Asp f1 allergen concentration in Bellaterra (Barcelona, Spain), during 2021 and 2022, and to compare their dynamics to improve the understanding of the biology of this genus and contribute to a better diagnosis, prevention, and therapeutic measures in front of possible health problems. The results show that both particles were airborne nearly all the year round, but their concentrations did not show a correlation. Due to Asp f1 is not being present in the conidia itself but is being detectable during their germination and in hyphal fragments, we report the relevance on of the aero-immunological analysis as a methodology to detect the potential pathogenic hazard of this fungus.

Keywords: Please consider organizing by alphabetic order.

Aerobiology; Asp f1; ELISA; fungal spores; High volume sampler; Hirst sampler

Introduction: Please consider replacing with:

Airborne fungal particles are an important essential fraction of atmospheric indoor and outdoor components that can be released by active or passive processes [1]. In general, fungi release large amounts of spores (sexual or asexual) and hyphal fragments into the air, which can be involved in the decomposition of food and raw materials, or the deterioration of organic matter during storage [2]. However, airborne fungi cause serious health problems, such as induced inflammatory responses or respiratory tract infections [3,4]. For example, in Europe, the incidence of fungus-induced respiratory tract allergies are as high as 20%-30% in the atopic population, reaching 6% in the general population [5]. Actually, More than 112 fungi fungal allergens are recognized as allergenic sources [6]. One of the most important is Aspergillus, considered the major primary fungal causative agent of allergic bronchopulmonary mycosis (ABPM) and respiratory allergic symptoms [7,8]. This Ascomycota is a filamentous saprophytic fungus widely distributed and associated with soil, decaying vegetation, or seeds. However, some species, such as A. fumigatus, are considered opportunistic pathogens of immunocompromised hosts. Their conidia (asexual spores) and mycelia are associated with clinical diseases related to the respiratory tract, such as allergic asthma, ABPM, or hypersensitivity, collectively termed ‘aspergillosis’ [9].

There are currently defined five major molecular allergens of A. fumigatus (Asp f1, Asp f2, Asp f3, Asp f4, and Asp f6). Among them, Asp f1 is the most important. Its production is related to the germination of spores and growth of the mycelia´s early fungal invasion [10], which is related to fungal colonization and the saprophytic nature of the fungi. Nowadays, there is no universal, recommended method for continuous air sampling that can accurately show the entire aeromycobiota of a location [11]. The Volumetric methods based on Hirst-type samplers offer a continuous sampling from the air with the reliable determination of fungal spores. However, it is less effective in the case of particles with a diameter below 5 µm, as occurs with Aspergillus, Penicillium, and some basidiospores, which may be underestimated as the number of spores in the air [11]. Another problem is the identification of fungal spores. In the case of Aspergillus and Penicillium, conidia of both species present similar morphology indistinguishable under the light microscope. Therefore, in aerobiological studies, these fungal species are grouped as Aspergillus/Penicillium spore type [12]. Moreover, this methodology cannot identify smaller fungal fragments and hyphae [13]. The viable methods (cultivation on the medium) can resolve the identification of these genera; they even can identify in-order to species [14]. However, continuous samples are not possible.

In contrast to these traditional methods, the immunological test may detect the presence of fungi based on the detection of their major allergen. Although this methodology depends on the commercial availability of specific allergens [13], it could be an alternative method for continuous airborne control of these fungi.

This study compares the airborne dynamics of Aspergillus/Penicillium spore type and the concentration of Asp f1 allergen in order to contribute to better diagnoses, prevention, and therapeutic measures for aspergillosis patients.

Material and methods: Please consider replacing with:

2.1. Sampling site

The aerobiological station is located at the Universitat Autònoma de Barcelona (Bellaterra, Barcelona, NE Spain), on the rooftop of the C building, at 23 m.a.g.l. (41°30’20’’N, 02°06’28’’E and 245 m.a.s.l.). The climate of this locality can be described as fresh (Mean annual Temperature 12.0–15.5 °C) and humid (Total annual Precipitation 400–700 mm), and the corresponding phytoclimate is Fresh-Continental Oriental-semihumid [15].

2.2. Aerobiological methodology

A Hirst-type volumetric spore trap (VPPS 2000 spore trap, Lanzoni s.r.l.) was used for the aerobiological fungal spore study. This collector aspires at a flow rate of 10L/min, comparable with the respiration of an average human adult, and was designed specifically for the intake of spores, pollen, and other particles suspended in the air [16]. It is a continuous volumetric sampler with wind orientation of the intake orifice, and, thanks to an aspiration system, the collected particles are deposited on a plastic band coated with silicone oil tightly deposited around a drum that is displaced at a rate of 2 mm/hour using a clockwise mechanism. Once a week, the plastic band in the drum is replaced, and the segments corresponding to each day (0-24 hours UTC) are mounted on a slide that is analyzed at the light microscope. The spore monitoring was performed continuously between 1st January 2021 and 31st December 2022 under the Spanish Aerobiological Network (Red Española de Aerobiología, REA) [17] and following the minimum requirements of the European Aerobiology Society, EAS [18]. The terminology used in this paper follows the International Association for Aerobiology (IAA) and EAS recommendations [19]. The data used in this study were expressed as the daily average of spores per cubic meter of air (spores /m3 ) and Annual Spores Integral (ASIn, spore*day/m3 ) or the addition of the average daily spores’ concentration over the year. The fungal taxa considered were Aspergillus/Penicillium spore type. In both cases, the asexual spores (conidia) extend out in long chains from the vegetative mycelium and readily detach to become airborne [20]. The size of this spore type is around 2.5 µm in diameter, with a spherical morphology that may have faint cinereous pigmentation or be hyaline. On their surface appear slight ornamentation similar to minute dots at the perimeter. Under a light microscope, it is frequently observed that the spores are in short chains or small groups adhered together (Figure 1). Due to the difficulties in the differentiation of this morphology and because the conidia may be hidden along with other airborne particles, it is very plausible that they are underestimated.

Added to the information generated during this study, the daily concentrations of airborne Aspergillus/Penicillium spore type in the Bellaterra sampling site for the ten years (2011-2020) were used to analyze the representativity of the results obtained for years 2021 and 2022. These data were extracted from the database of the Catalan Aerobiological Network (Xarxa Aerobiològica de Catalunya, XAC) and are summarized in Table 1.

2.3. Immunochemical methodology

Besides the Hirst sampler, the Asp f1 sampling was performed during 2021 and 2022, using a high-volume sampler (MCV CAV-A/mb, © MCV) designed in principle for the sampling of atmospheric particulate matter. It is a volumetric sampler, working at a flow intake rate of 400 L/min and impelling the air against a fibreglass filter (UNE- EN12341:2015; UNE-EN 14907:2006a, b) that will retain the particles. Based on MCV S.L. Company, this collector complies with the specifications for high-volume collectors in the UNE-EN 12341 and UNE-EN 14907 Standards. The total diameter of the filter is 15 centimetres, but the diameter of the impacted area is 12 centimetres. The daily samples were carried out from 0 to 24 hours, and the sampling was carried out on alternate days to permit manual replacement. The samples were stored at 2°C until their analysis. Before the analysis, a representative part of the daily filters, consisting of 8 circles of 0.5 cm in diameter obtained in selected areas, was equally distributed in two Eppendorf vials with 500 µl PBS solution and hydrated for 6 hours [21]. Once hydrated, the samples obtained were analyzed according to the commercial ELISA kits to detect Asp f 1 (Inbio, Indoor Biotechnologies, Inc, Cardiff, United Kingdom) according to the protocol provided on the supplier’s website [22]. The allergen standards used were provided with the kits. The microplates used were NUNC™ microplates. Absorbance measurements were made at a 492 nm filter in a Plate reader model Halo MpR-96 from Dynamica® . The results were expressed as daily allergen concentrations in nanograms of allergen per cubic meter of air (ng/m3 ) and as Annual Allergen Integral (AAIn) obtained by summing the average daily allergen concentration over the year (ng*day/m3 ).

2.4. Meteorological data

The meteorological variables considered in this study were maximum and minimum daily temperatures (expressed in Celsius degrees), daily rainfall (expressed in millimetres), and relative humidity (expressed in percentages). Data were obtained from the Servei Meteorològic de Catalunya and correspond to the Sabadell meteorological station (41°33’56´´N, 02°04’10´´E, 258 m.a.s.l.).

2.5. Statistical analysis

Spearman’s correlation coefficients were calculated between the daily concentrations of Aspergillus/Penicillium spore type and Asp f1 allergen and daily meteorological variables. The analyses were carried out by using the IBM SPSS Statistic version 25.

Results: Please provide a re-written version of this section.

Discussion: Please consider replacing with:

The study of the daily dynamics of the Aspergillus/Penicillium spore type along the ten years previous to the analysis of Asp f1 (Table 1, Figure 2) has evidenced that it does not follow a clear seasonal pattern, that the spores are observed in the aerobiological plates a critical number of days per year (between 50% and 72% of the days analyzed per year) but not continuously, and that they are distributed over several months. However, the maximum airborne spore concentration period in autumn (October and November) can be intuitively established. This yearly phenomenon has been reported by other studies, such as Oliveira et al. [23] in Portugal and Rosas et al. [24] in Mexico. However, the dynamics of this spore type around the world are variable. Several authors have shown that the main period occurs during spring, as in Central Spain, Salamanca [25]. In contrast, others reported a two-season occurrence (spring and autumn) as in Poland [11] or in Ireland [26]. In the USA [27] and Finland [28]the main period was observed during summer. Finally, some studies demonstrated that Aspergillus/Penicillium spore type is independent of seasonal changes, such as in Denmark [29]. Although the period of high fungal spore concentrations does not show a unique pattern, it is indisputable that the existence of this spore type in the air has implications for patients with problems in the respiratory system [20]. Even so, there is no general agreement on analyzing the presence of airborne fungal particles in the atmosphere. Some authors analyzed the air using viable sampling methods that resulted in the calculation of colony-forming units (CFU), which means brief periods of sampling and sporadically through the year [30-32]. Other authors use aerobiological methods based on palynology (with the identification of the particles through an optical microscope) [23,25]. In this case, the air sampling was continuous, but there needs to be a consensus on the surface of the sample that will be analyzed. These aerobiological studies have a substantial limitation in counting and identifying the numerous and diverse fungal spore types present in the atmosphere, especially in the case of Aspergillus/Penicillium spore type, due to their small size and hyaline colour. Furthermore, this methodology cannot identify smaller fungal fragments or hyphae, considered a significant source of allergens [33], so the actual allergenic risk due to Aspergillus would still not be observed.

In the last decades, aerobiological studies have evolved, incorporating molecular analytical methods, such as detecting and quantifying the allergen load in the atmosphere through immunological techniques. This new methodology allows the comparison of their results with those from traditional sampling, achieving a better knowledge of the natural aerobiological dynamics of pollen, spores, and allergens. Therefore, the information obtained with both methodologies can help assess pollen/spore exposure and improve research on epidemiology and clinical trials. The present study offers an analysis of the airborne Aspergillus/Penicillium spore type and the allergenic load of Asp f 1 in Bellaterra (Barcelona) to determine if the traditional aerobiological method reflects the actual risk period of Asp f1 in the air. 

This study recorded airborne dynamics of Aspergillus/Penicillium spore type in 2021 and 2022. As shown in our results and other research [34-35], the levels of this spore type can be variable throughout the year, with high values in some seasons and low values in others. During the period studied, high daily concentrations of spores were registered on certain days in the year exceeding 200 spore/m3. While in 2021, this occurred on some days of April, May, and June. In 2022 it was registered in May, June, and October. This fact, and the lack of correlation with meteorological variables, could be explained by the sporadic behaviour of these spores [11]. However, the positive and significant correlation between the two years studied (0.145; p< 0.01) shows how this spore type presents similar annual airborne dynamicsalthough some days randomly register high concentrations. 

In this study, the authors used a high-volume sampler with a suction capacity of 400 L/min. The election of this sampler was based on the hypothesis that this high-volume collector could collect biological particles, such as allergens, at the same time as inorganic particles. As shown in De Linares et al. [21], sample collection analysis and quantification of fungal allergens are possible with this sampler and using the ELISA technique.

Despite several studies on airborne allergens, they are mainly focused on allergens from pollen, and there need to be more studies on fungal spores. The research of aeroallergens from fungal spores was mainly based on the detection and quantification of Alt a 1 [21,36-37] because it is the most critical fungal aeroallergen that provokes respiratory tract symptoms. Studies related to outdoor Aspergillus spore and their major allergen (Asp f 1) are scarce. In this case, we found one article using a similar methodology: a volumetric sampler with a flow rate of 40 L/min [38] to quantify the airborne Asp f1. As in this study, we found that the seasonal distribution of airborne spores did not correlate with the aeroallergens load. In our case, during the two years study (2021-2022), the higher airborne spores’ concentrations were registered in spring and autumn, the peak days on 23 May and 28 May, respectively. The fact that the maximum spore concentrations were registered in the more humid season agrees with other studies [11,34,35,38,39]. However, the maximum concentration of the Asp f1 allergen was found during winter, peaking

on 18 February and 11 February, as occurred in Vermani et al. [33], which detected the highest outdoor Asp f1 concentration of A. fumigatus and A. niger in February too. 

Statistical results showed no significant correlation between meteorology and Aspergillus/Penicillium spore type. Only in 2021 were positive correlations registered with Precipitation and Relative Humidity. The same lack of correlation with temperature was obtained in Grinn-Gofron [11], Sadys et al. [40], Akgül et al [41]. In the case of Asp f1 allergen, non-correlation was detected in any year, so it could indicate that the weather conditions do not influence these small particles and depend on other mechanisms that must be studied.

Consequently, based on the discrepancies in the moments of these two particles being present in the air and the similitudes on the date of maximum concentrations in both years and with other studies, we postulate that: a) the coincidence of the values of ASIn, AAIn and dates of the peaks of spores and allergens, as well as the dynamic found during the two years studied, confirm that the detection of the Aspergillus/Penicillium spore type and Asp f1 allergen is possible with both aerobiological techniques, and follow a defined behaviour along the time. 

b) if we accept the postulate of Sporik et al. [30] that Asp f1 is not present in the spore but in hyphae fragments, and as previous studies have shown, we accept too that the level of the fungal hyphae can be relatively high in the air [42].

c) Aspergillus/Penicillium spore type is conditioned to precipitation and relative humidity, which promote the production and release of the spores of these fungi. On the other hand, the aeroallergens registered in winter could be derived from hyphal fragments, which, under the light microscope, are extremely difficult to distinguish but can be detected through the allergens with immunological techniques.

Conclusions: Please consider replacing with:

Asp f1 daily concentration measurements have been obtained, for the first time in Catalonia, at the Bellaterra (Barcelona) aerobiological station, for the years 2021 and 2022, showing that the MCV CAV-A/mb high-volume sampler has proven to be valuable and sensitive enough to the detection and quantification of the allergen Asp f1 using ELISA Kits.

Despite being unable to explain specific temporal differences observed between concentrations of spores and allergens, we recommend using these two methodologies (quantification of fungal spores under a light microscope and immunochemical quantification to detect allergens) to detect, control, prevent and diagnose this fungal taxon. Further studies must focus on establishing the origin of the aeroallergen registered in periods without airborne fungal spores and determine at which moment is more hazard the exposure when there are high concentrations of spores or the allergen. 

Scientific comments:

Have the authors considered to use moving averages (for instance 3 to 5 days) for Asp/Pen spore concentration and then compare it Asp f1 allergens?

English quality needs to be improved.

Author Response

We want to thank the reviewer for the work done, which help us to improve the manuscript. We have accepted most of the proposals marked in bold and have taken the comments into consideration. You can see it in the manuscript with the changes marked and in the responses to reviewer 2.pdf

Reviewer 3 Report

please see file attached. 

please see file attached

Author Response

We want to thank the reviewer for the work done, which helps us to improve the manuscript. We have taken the proposals into account. You can see it in the manuscript with the changes marked and in the responses to reviewer 3.pdf

Round 2

Reviewer 2 Report

The reviewer acknowledges the authors´ efforts to improve the overall quality of the manuscript.

In the legend of Table 2 replace "Prec: annual rainfall" by "R: annual rainfall". Precipitation is more than rainfall.

Reviewer 3 Report

I have no further comments